# Automatic Essay Evaluation Technologies in Chinese Writing—A Systematic Literature Review

Hongwu Yang [1,*,†], Yanshan He [1,2,†], Xiaolong Bu [1], Hongwen Xu [1] and Weitong Guo [1]

1 School of Educational Technology, Northwest Normal University, Lanzhou 730070, China; heyanshan@mail.lzjtu.cn (Y.H.); bbbxxxlll@nwnu.edu.cn (X.B.); 2018221777@nwnu.edu.cn (H.X.); guowt@nwnu.edu.cn (W.G.)
2 School of Electronic and Information Engineering, Lanzhou Jiaotong University, Lanzhou 730070, China
* Correspondence: yanghw@nwnu.edu.cn
† These authors contributed equally to this work.

**Abstract:** Automatic essay evaluation, an essential application of natural language processing (NLP) technology in education, has been increasingly employed in writing instruction and language proficiency assessment. Because automatic Chinese Essay Evaluation (ACEE) has made some breakthroughs due to the rapid development of upstream Chinese NLP technology, many evaluation tools have been applied in teaching practice and high-risk evaluation processes. However, the development of ACEE is still in its early stages, with many technical bottlenecks and challenges. This paper systematically explores the current research status of corpus construction, feature engineering, and scoring models in ACEE through literature to provide a technical perspective for stakeholders in the ACEE research field. Literature research has shown that constructing the ACEE public corpus is insufficient and lacks an effective platform to promote the development of ACEE research. Various shallow and deep features can be extracted using statistical and NLP techniques in ACEE. However, there are still substantial limitations in extracting grammatical errors and features related to syntax and traditional Chinese Literary style. For the construction of scoring models, existing studies have shown that traditional machine learning and deep learning methods each have advantages in different corpora and feature selections. The deep learning model, which exhibits strong adaptability and multi-task joint learning potential, has broader development space regarding model scalability.

**Keywords:** automated essay evaluation; Chinese writing; natural language process; systematic literature review

## 1. Introduction

Automated Essay Evaluation (AEE) [1] refers to the technology of using computer programs to evaluate essays. Initially conceived to alleviate the workload for teachers and reduce grading subjectivity, AEE has evolved into an essential tool for educators and learners due to its ability to generate personalized feedback and recommendations.

The prevailing AEE methodologies aim to emulate human grading by employing computational linguistics methods to extract linguistic elements as evaluation features. This process involves data collection, text preprocessing, feature extraction, scoring model construction, and generating evaluation results, as shown in Figure 1. The core tasks are feature extraction and scoring model construction.

The research and application of AEE can be traced back to the 1960s with the development of the Project Essay Grader (PEG) system [2]. Over the years, features have evolved from shallow linguistic to deep linguistic elements [3–8] and evaluation methods have transitioned from traditional machine learning to neural network-based deep learning [9–13]. Moreover, research in English AEE has followed a well-established development pattern supported by multiple publicly available corpora and research platforms. In contrast, AEE

research in other languages has been slower due to the lack of research corpora and natural language processing (NLP) tools [14–17].

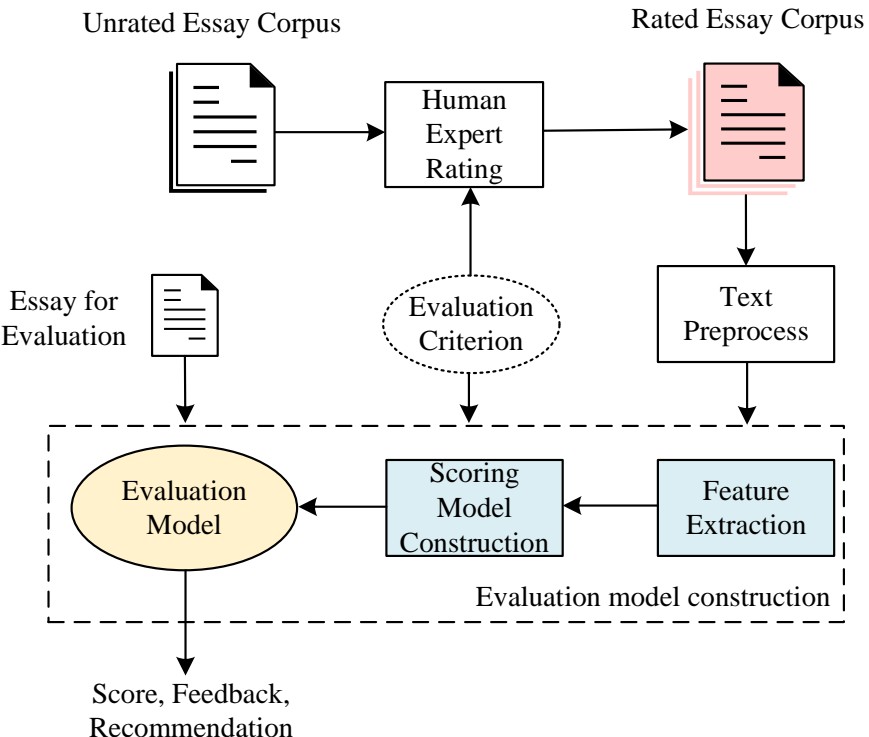

**Figure 1.** Process of AEE.

There is a rising desire for automated essay evaluation in Chinese, which has over a billion native speakers and a growing number of foreign learners. Inspired by research on AEE in English, Chinese scholars initiated Automated Chinese Essay Evaluation (ACEE) in the early 21st century [18–21]. Over the past two decades, this field has witnessed notable research achievements and accumulated valuable research experience. Multiple ACEE products have been applied in practice. Nonetheless, empirical studies [22–25] suggest that the limitation of ACEE technologies affects its effectiveness in writing teaching, and the lack of transparency raises skepticism regarding its validity and reliability.

Knowing the current state of ACEE technology is crucial for informed decision-making by researchers, developers, and stakeholders and encourages more engagement from professionals in the field. Several studies have provided comprehensive reviews of the existing studies on ACEE from different periods and perspectives. Liu, W. et al. [26] classified essay evaluation as subjective question evaluation with open-ended answers and reviewed some early research findings of ACEE. On the other hand, the studies conducted by Wu, J. et al. [27] and Rong, W. et al. [28] provided varying degrees of comprehensive reviews of ACEE's key technologies and practical applications, offering insights into its future development from the perspectives of technology, research platforms, and educational practices. Additionally, Xue, S. et al. [29] focused their research on the advancement of ACEE technologies, with evaluations conducted on two distinct groups: native Chinese speakers and non-native Chinese speakers (abbreviated as NS and NNS in the following content). However, the discussion of ACEE technologies is not exhaustive due to limitations in the research duration and the literature sources, so there has been a shortage of detailed comparative analyses of existing research efforts.

The primary objective of our study is to conduct a systematic literature review (SLR) for the benefit of researchers, technicians, and other stakeholders involved in ACEE. This review aims to provide the most exhaustive knowledge in a technically rigorous manner and to identify potential directions for future research. SLR, a comprehensive and method-

ologically rigorous approach to reviewing existing research findings, is widely used in academic and scientific research to inform evidence-based decision-making and to identify gaps in the existing literature for further investigation. Kitchenham, B. et al. [30–32] introduced typical research stages for SLR and demonstrated the typical research framework of SLR in several papers.

According to the framework proposed by Kitchenham, B. et al. [31], the SLR in this article consists of the following specific processes. Initially, a review protocol needs to be developed, including research questions, literature sifting rules, and analysis methods. According to the protocol, a systematic search and selection process was conducted as a beginning to identify academic journals and conference articles that reflect the advancements in ACEE technology research as the basis for the review. Subsequently, we focused on the core tasks within the automated essay scoring process with qualitative and quantitative analysis methods to comprehensively review the research progress in three key processes: corpus construction, feature extraction, and evaluation model development. Lastly, we explored the future directions for ACEE studies by analyzing the deficiencies and challenges present in ACEE research.

## 2. Materials and Methods

### 2.1. Research Question

This study aims to answer the following questions according to the crucial technological issues that need to be addressed in each key stage of the automated essay-scoring process.

- RQ1. What is the status of construction and the usage of corpora in ACEE research?
- RQ2. What features can ACEE extract for essay evaluation, and what are the key techniques and methods involved in the extraction process?
- RQ3. What are the key technologies and methods for constructing the ACEE scoring model?

### 2.2. Literature Collection and Selection

To comprehensively obtain relevant research literature on ACEE while ensuring the quality of the literature, we conducted a literature collection and selection process through four steps: database search, backward search, filtering with inclusion and exclusion criteria, and filtering with a quality assessment, as shown in Figure 2.

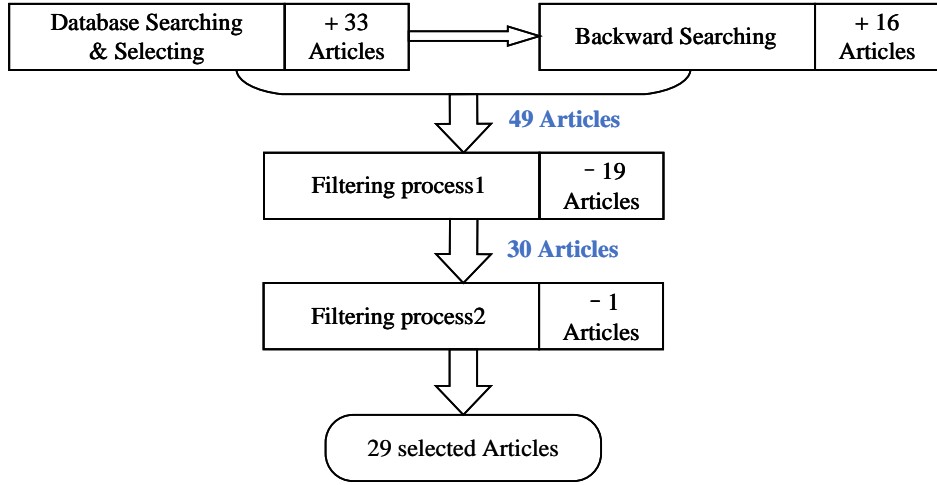

**Figure 2.** Process of literature collection and selection.

### 2.2.1. Search Process

Two methods, database search and backward search, were employed to collect literature. Firstly, we searched the widely recognized academic search platforms, including IEEE Explore, Web of Science, Science Direct, Engineering Village, and CNKI, with the keywords ('automated') ('essay' or 'writing' or 'composition'), and ('grading' or 'scoring' or 'evaluation') to obtain related articles. Because we found that most ACEE literature does not explicitly mention the language in titles, abstracts, and keywords, we did not add 'Chinese' to the search keywords. We selected ACEE-related literature by reviewing the abstracts from the search results according to the following criteria.

1.  The literature study automated Chinese composition evaluation with the Chinese corpus and does not include any research on other languages.
2.  The literature includes only journal articles or conference papers and does not include theses, reports, etc.

Next, we used a backward search to explore the references within the selected literature to identify additional relevant papers that meet the abovementioned criteria. This process was iterated for multiple rounds until no new literature was found.

### 2.2.2. Filter Process

For the collected literature, two rounds of screening were conducted by carefully reading the details of each paper. We filtered the selected literature based on the inclusion and exclusion criteria in Table 1.

**Table 1.** Inclusion and exclusion criteria.

| Inclusion Criteria | Exclusion Criteria |
| --- | --- |
| IC-1. The research content of the literature focuses on ACEE technologies. | EC-1. Research on the practical application of ACEE and review literature. |
| IC-2. The research revolves around automated essay scoring as its primary task or includes relevant research content on essay grading. | EC-2. Literature solely dedicated to studying upstream NLP tasks related to ACEE. |
| IC-3. Translation: Literature needs to have a certain level of academic influence, primarily demonstrated by being widely recognized and indexed by prestigious academic citation index or databases such as SCIE, SSCI, EI, CPCI, Core Journals of China ①, CSCD ②, or by having a high citation frequency within the respective field. | |

① A Guide to The Core Journals of China results from a research project led by Peking University on evaluating Chinese core journals, which are currently widely recognized by the Chinese academic community. ② CSCD is the abbreviation of the Chinese Science Citation Database and is known as the "SCI of China".

Through the above filter process, we selected 30 articles. Figure 3 shows the citation status of literature chosen by SCIE/SSCI, EI/CPI, and CSCD/Core Journals of China. We can see that three articles are not included in the databases or catalogs mentioned in Table 1. However, these articles were cited 69, 31, and 27 times, respectively (from CNKI), with the citation frequency ranking in the top 20%. So, we included these three articles in the review.

### 2.2.3. Quality Assessment

We conducted a quality assessment of the selected literature to reduce potential bias during the search and selection process. The assessment was based on the following quality assessment questions by adopting the framework proposed by Kitchenham, B et al. [31].

*   QA1: Quality of the corpus.
*   QA2: Innovation in research methods and technology.
*   QA3: Evaluation of feature effectiveness.

- QA4: Evaluation of the essay scoring model.

Based on each quality assessment question, the literature was rated according to three scores: 2 (Well done), 1 (Medium), and 0 (Not mentioned). The final quality evaluation score of the literature is obtained by summing the scores of the four questions mentioned above. The maximum score is 8 points. Articles with scores below four are excluded during this process. Two researchers conducted the evaluation process through peer review. The average Quadratic Weighted Kappa shows that the two researchers' agreement was 0.7685. Table 2 shows the literature's quality evaluation results. The quality assessment process excluded 1 article, leaving a final list of 29 selected papers.

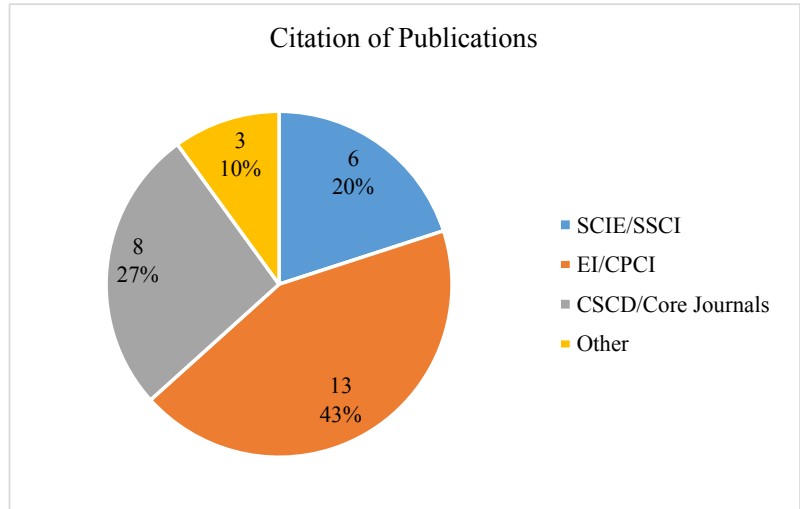

**Figure 3.** Cited publications.

**Table 2.** Results of quality assessment.

| Number of Papers | Quality Assessment Score |
|:---:|:---:|
| 9 | 7~8 |
| 20 | 4~6 |
| 1 | 0~3 |

### 2.3. Data Extraction

Regarding the research question proposed in Section 2.1, the researchers thoroughly reviewed the selected literature and extracted information related to corpus construction, feature extraction, and scoring model development, including the source and size of the corpus, essay topics within the corpus (whether they are single-topic or multi-topic), the extracted or selected features or feature sets, the method of feature extraction, the approach to constructing the scoring model, the evaluation method, and the evaluation results.

Various features, feature extraction methods, and scoring model constructions are categorized into several major classes to facilitate statistical analysis and comparison. All the data are stored in a single Excel sheet. Subsequently, researchers classified and aggregated the data from the source table into multiple analysis tables based on specific requirements. Finally, the results were presented and analyzed using charts and tables.

### 3. Results

### 3.1. Corpus Construction and Usage (RQ1)

The corpus serves as the foundation for AEE research, so we depicted the usage of corpora in the ACEE research literature in Figure 4. Some scholars utilized large-scale exam corpora such as HSK (Hanyu Shuiping Kaoshi, the Chinese Language Proficiency Test) [18,33–36] and MHK (Minzu Hanyu Kaoshi, Chinese Language Proficiency Test for

Ethnic Minorities in China) [19,37–41]. The HSK and MHK corpora contain essays written by NNS for Chinese proficiency assessments. In contrast, most of these corpora consisted of compositions from local primary and secondary school students, typically considered NS.

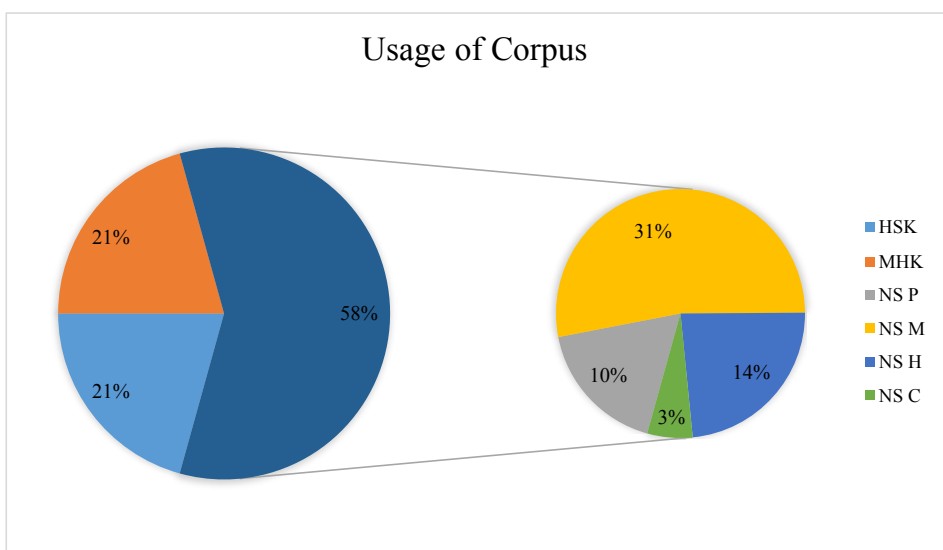

**Figure 4.** Usage of corpus.

"HSK Dynamic Composition Corpus" http://hsk.blcu.edu.cn/ (accessed on 2 September 2023), constructed and maintained by Beijing Language and Culture University, is the only publicly available large-scale Chinese essay corpus in mainland China. The newest version of the corpus is v2.0, which collects essay responses from non-native Chinese speakers who participated in the HSK examination between 1992 and 2005 and contains 11,569 essays, amounting to 4.24 million characters. Errors found in the corpus have been categorized and annotated from the perspectives of characters, words, sentences, paragraphs, and punctuation. Two professional assessors score the essays in the HSK corpus, and the average score is considered as the candidate's essay score, ranging from 40 to 95.

MHK is a national standardized examination designed to assess the Chinese language proficiency of non-native Chinese speakers from ethnic minority backgrounds. The human scoring of the MHK corpus has high levels of reliability and validity. However, this corpus is not available to the general public, making it difficult to obtain.

The vast majority of self-built corpora in research have not been disclosed in any way on any platform, and only Li, H. et al. [33] have disclosed the corpora used in the study on Github, including 300 essay texts and rating information used in the study. However, the grading details of the corpus were not mentioned, so the quality of the corpus cannot be evaluated.

*3.2. Feature Extraction and Selection (RQ2)*

Feature extraction, which plays a significant role in automated essay scoring by transforming the essay into quantifiable features, involves extracting various features or indicators from the essay text that contribute to evaluating the quality of the essay. These features and indicators can be based on linguistic, semantic, structural, and contextual aspects. In addition to being used in constructing essay evaluation models, these features and indicators serve as the basis for providing reliable feedback and improvement suggestions. Commonly extracted features in ACEE include length features, lexical features, spelling errors, grammatical features, syntactic features, thematic features, discourse features, rhetorical features, etc. Apart from these independently meaningful features, text embedding representations retain vast textual information, making it convenient for deep neural network models to extract features automatically. Consequently, in some studies, they are also called embedding features. This section provides a detailed overview

of the techniques, methods, and tools used in constructing and extracting the features mentioned above.

### 3.2.1. Length-Based Features

The length-based features are a typical kind of statistical feature often reflected in the evaluation standards in most high-stakes exams and daily writing tasks. These features include the total number of words, the total number of sentences, and the number of advanced-level vocabulary words [18,19,33]. They generally indicate the fluency of expression in terms of form and have varying degrees of impact on the evaluation of essays in various types and levels of writing.

### 3.2.2. Lexical Features

The difficulty level of vocabulary used in the text is a commonly used feature to evaluate the level of writing. Especially in evaluating Chinese non-native language compositions, the number and frequency of vocabulary appearing in the text [18,19,33] at or above the second difficulty level (Yi) (in this article, we classify vocabulary into four difficulty levels based on the "Chinese Proficiency Vocabulary and Chinese Character Level Outline" developed by the National Office for Teaching Chinese as a Foreign Language. The four Chinese vocabulary difficulty levels are the first level (Jia), second level (Yi), third level (Bing), and fourth level (Ding), ranging from easy to difficult) are often used as characteristics to evaluate language proficiency. Cai, L. et al. [37] evaluate the writing proficiency of compositions by computing the difficulty coefficients of vocabulary used within the text. In addition to vocabulary difficulty levels, rhythmic words are a distinctive feature of written Chinese expression that can enhance the quality of compositions. Feng, S. et al. [42] have researched the rhythmic features of written language and explored the correlation between the elegance of compositions and certain rhythmic patterns, such as embedded paired monosyllabic words (rhythmic modules consisting of paired classical monosyllabic words), coupled disyllabic words (language rhythmic modules appearing in pairs), as well as the validity of archaic functional words and vocabulary of the fourth level (Ding) in composition scoring. The experimental results confirm a significant positive correlation between the elegance feature and the level of compositions.

The Chinese version of Coh Metrix and text analysis tools, such as CCRL and ICTCLAS, developed in Beiyu can quickly and effectively obtain length and basic vocabulary features, greatly facilitating ACEE research.

### 3.2.3. Spelling Errors

In contrast to English, Chinese characters and words have unique rules and characteristics regarding construction and usage, often requiring contextual understanding. Chinese spelling errors can be categorized into two types: "cuozi" (wrong character) and "biezi" (confusing character). "Cuozi" refers to the characters that do not appear in the Chinese character library due to stroke or radical errors during hand-writing. Identifying "cuozi" in Chinese compositions relies on techniques such as optical character recognition and evaluation, which have not been extensively investigated in existing ACEE research. "Biezi" refers to incorrectly using characters due to similar pronunciations or shapes. Early studies relied on manual recognition and annotation [18,19]. Researchers have recently established corpora or confusion sets of "biezi" to automatically recognize and annotate them using pattern matching or language models. Yang, Y. et al. [43] constructed a "biezi" recognition corpus, including customized corpora from elementary textbooks, idiomatic expressions with high "biezi" occurrence rates, and normal corpora from newspaper texts. They also developed detection rules based on the corpus to identify "biezi" in the text. Hao, S. et al. [44] established a table of easily confused characters and used WFSA to detect "biezi" in compositions using an n-gram model while creating a table of commonly misspelled characters for correction. Wei, S. et al. [45] labeled the positions of "biezi" and provided correction suggestions by combining the Soft-Masked BERT (Bidirectional

Encoder Representations from Transformers) [46] model with tables of phonetically and visually similar characters.

### 3.2.4. Grammatical Errors

Common grammatical errors in Chinese writing include missing or redundant sentence components and inappropriate collocations. The HSK corpus has been manually annotated to identify these common grammatical error types in Chinese writing. The types and quantities of annotated errors [18,19,33] were used as evaluation indicators in the early stages of the ACEE study. In recent years, research has focused on automatically annotating grammatical errors using deep learning-based sequence labeling tasks. The current technology can detect errors such as word order disruptions, redundant or missing words, and more. Wei, S. et al. [45] introduced the grammar error detection module in the iFLYTEK ACEE system (known as IFlyEA), which utilized a multi-layered bidirectional transformer encoder model integrated with ResNet for sequence labeling tasks to annotate the types and positions of grammatical errors within the essay. This method achieved the highest F1 score in error localization and recognition tasks in the Chinese Grammatical Error Diagnosis (CGED) evaluation competition in 2022. Yang, Y. et al. [43] used a Bi-LSTM-CRF model that won first place in the CGED 2021 to detect and annotate grammatical errors.

### 3.2.5. Syntactic Features

In ACEE, syntactic features primarily refer to the complexity of syntax. This feature plays a vital role in evaluating the writing quality of second language learners, as proficient authors often use more complex sentence structures in Chinese writing. Syntactic complexity can be measured by sentence feature-based indicators such as average sentence length [18,19], the number and length of long sentences, and T-units [35,47]. In recent years, more sophisticated syntactic features have replaced sentence-level features to improve the interpretability and effectiveness of evaluation. For example, average syntactic tree depth, the number of complex semantic units, and the complexity of grammatical structures have been used to measure the syntactic complexity in compositions. Wang et al. [35] utilized the minimum T-unit length, average and maximum syntactic tree depth, as well as the density and ratio of grammatical structures (annotated and extracted from the "International Curriculum for Chinese Language Education" [48]) to indicate the syntactic complexity in second language writing. Dependency distance is an important measure of syntactic complexity in dependency grammar models. Yang et al. [49] represented the conceptual relationships in an article using concept graphs and calculated the average dependency distance between concepts to reflect the syntactic complexity of compositions.

### 3.2.6. Thematic Features

The requirement for an essay to be "relevant to the topic, with a clear theme and prominent central idea" is expressed in evaluating the thematic concept in Chinese writing. In ACEE, this is represented by theme-relevant features and semantic coherence features.

In ACEE, there are primarily two methods for extracting theme-relevant features. One way is to compute the semantic relevance between the prompt or topic and the essay under evaluation. In the study of Li, H. et al. [50], the thematic description was treated as a long sentence input to the BERT model to obtain word vector representations related to the theme. This information was then used as attention information to enhance the theme relevance in the text representations generated by the DNN (deep neural network) at each layer. Sun, J. et al. [34] obtained theme relevance features through two classification tasks: theme prediction and theme matching. Theme prediction involves predicting the theme that an essay belongs to from a limited set of themes, while theme matching determines whether the essay is compatible with a specific theme. The study used a deep neural network with multitasking optimization to achieve the highest F1 values of 98.1% and 91.4% in the two tasks mentioned. Another way is to calculate the relationship between the themes of the essay to be evaluated and those presented in the corpus. Cai, L. et al. [37]

utilized word frequency techniques to extract the theme features from a corpus, where the ratio of word frequency in the test essay to it in the corpus was used as the coefficient of theme relevance for that word. The final theme feature value for an essay was the sum of theme relevance coefficients for all words in the essay. This method improved upon measuring theme relevance by computing the similarity between the essay title and the word vectors of the test essay. Hao, S. et al. [40] adopted regularized latent semantic indexing (LSI), a theme model, to extract theme features. A term-theme matrix and a theme–document matrix are generated from a TF-IDF (term frequency-inverse document frequency) weighted vocabulary–document matrix. By training the theme–document matrix, a theme vector space is formed. The word vectors of the test essay are then mapped onto this theme vector space to obtain theme-based text vectors.

The theme specificity measurement is quantified through textual semantics' coherence features. Wang, Y. et al. [51] mapped the essay and each sentence within it into a unified latent semantic space, which can be represented as Latent Dirichlet Allocation (LDA) or Word Vector with TF-IDF values. Then, the distance vector between each sentence vector and the essay vector was computed to extract the semantic distinctiveness that indicated the theme relevance of the essay.

The research conducted by Yang, L. et al. [49,51] adopted multiple levels of tokens as conceptual nodes by constructing a concept graph of essays with the dependency relationships among concepts and the semantic distances between them. Finally, they extracted several features, including global convergence, local convergence, distance between nodes, and similarity to the high-scoring essays' concept maps, to obtain the thematic characteristics of the essays. These methods enable the simultaneous evaluation of theme relevance, salience, and theme development in the essays.

### 3.2.7. Discourse Features

Discourse features exhibit the organizational structure of an essay and reflect the author's logical reasoning abilities. Research on discourse features in the context of ACEE includes intra-paragraph and inter-paragraph structural features. The main methods are based on lexical chains, position, and discourse elements.

The lexical chains-based approach extracts discourse information based on the relationships among words or concepts (a collection of semantically similar words). In the study of Yang, L. et al. [52], the intra-paragraph discourse structure was represented using a sequence structure of concepts within the paragraph. The study first employed a concept hierarchy model to represent a set of concepts within the paragraph. The order of appearance of these concepts within the paragraph and the relational information between them were utilized to construct the sequence structure of concepts within the paragraph. Finally, the similarity between the concept structures within the tested essay and those within the training set essays was compared to assess the rationality of discourse structure.

The position-based approach utilizes the direct or relative positional information of sentences or paragraphs as discourse features. Liu, J. et al. [53] determined the logical coherence within essay paragraphs by assessing the reasonableness of sentence order. They represented paragraphs as sequences of sentences and input them into multiple classification models to evaluate the logical coherence within paragraphs. Experimental results demonstrated that utilizing the initial, final, or combination of both as sentence features achieves the highest discrimination accuracy when using a Bi-RNN classifier capable of effectively capturing the connecting information between adjacent sentences.

The discourse elements refer to the functional roles of sentences and paragraphs within an essay. Basic characteristics of organizational structure can be derived by identifying these discourse elements. This approach typically requires a sufficiently large annotated training dataset of discourse elements. Song, W. et al. [54] employed a two-layer Bi-LSTM model to extract features of different hierarchical elements in the composition and constructed a two-dimensional CNN model to evaluate the organizational structure of the essay.

### 3.2.8. Rhetorical Features

Literary style plays an essential role in evaluating Chinese writing in assessing the content. The manifestation of literary style mainly includes using rhetorical devices and incorporating poetry, famous quotations, and allusions. Commonly used rhetorical devices in Chinese writing include metaphor, personification, parallelism, and exaggeration. These rhetorical devices often possess distinct structural features. Therefore, rule-based methods, pattern matching, and data-driven approaches are employed for extracting rhetorical features.

The simile is a special type of metaphor. It is observed that simile often possesses specific text formats and recognizable markers. The research [43] on extracting shallow features with statistical methods identifies metaphorical sentences by detecting metaphorical words. However, this method is unreliable because Chinese sentences containing metaphorical words are not necessarily metaphorical sentences. For instance, the Chinese sentence "他好像根本不知道这件事一样" ("He seems not to know this matter at all") may contain metaphorical words but is not a metaphorical sentence. Recent studies have improved the identification of sentences with similes by incorporating organizational structural features. In the study of Chang, T. et al. [20], typical patterns of lexical chains in metaphorical sentences were identified based on the part-of-speech organization after identifying potential metaphorical sentences through the recognition of metaphorical words. Given the diverse syntactic forms of metaphorical sentences in Chinese, Liu, L. et al. [55] constructed a multi-task end-to-end neural network model to learn word sequence features in simile sentences for both identifying simile sentences and recognizing their constituents (the tenor and vehicle) in the tested essay.

Parallelism combines multiple clauses or phrases within a sentence with similar syntactic structures or expressions. Automatic recognition methods for parallelism sentences are often designed based on their unique features in structure and expression. Chang, T. et al. [20] identified the similarity in structures of clauses within parallelism sentences by segmenting the sentence and performing part-of-speech tagging on the resulting sentence structure. Song, W. et al. [56] identified sentence parallelism by detecting features such as word alignment, semantic alignment, and sentence structure alignment. The study obtained word alignment features by checking if the characters, part-of-speech, and grammatical roles of words in clauses match. Word embedding techniques are used to calculate semantic similarity between words to obtain semantic alignment features. Alignment features for sentence structures were obtained by identifying the longest common subsequence and measuring the similarity of syntax trees.

Chinese composition can add depth and sophistication to the writing, making it more expressive and impactful using idioms, poems, proverbs, and allusions in quotations. In ACEE, information retrieval and pattern-matching techniques are used to recognize citation sentences on existing or self-built citation resource databases [57,58].

Elegant sentences always serve as one of the bonus criteria in the scoring standards for Chinese writing. However, ACEE research currently has no unified definition of elegant sentences. Some researchers consider sentences with rhetorical devices elegant, while others believe that elegant sentences possess certain implicit characteristics. Fu, R. et al. [45,59] consulted the criteria for evaluating elegant expressions in the Chinese College Entrance Examination (Gaokao) and suggested that eloquent sentences usually demonstrate outstanding performance in language, sentence structure, rhetoric, and quotations. This study determined the degree of elegance in sentences based on their content, expression style, and other factors. A hybrid DNN model based on CNN and BiLSTM was proposed to automatically identify elegant sentences using a data-driven and multi-task joint approach.

### 3.2.9. Embedding-Based Feature

The embedding technology maps textual data into a fixed-length vector space as informative representations of the text while preserving certain semantic information from the original text. These informative representations can be utilized for feature extraction or

directly inputted into machine learning models such as SVM and Deep Neural Network (DNN) to predict essay scores. In some studies, text embeddings are also referred to as "embedding-based features".

The Vector Space Model (VSM) [41] is an early form of the embedding model. Each vector dimension corresponds to a word, and the value represents the frequency of the word occurrence in the text (usually represented by TF-IDF). Latent Semantic Analysis (LSA) [44], also known as Latent Semantic Indexing (LSI), is an improvement upon the traditional VSM. LSA utilizes singular value decomposition to transform the Bag-of-Words (BoW) vector into a lower-dimensional space called the Latent Semantic Vector Space (LS-VS). LS-VS represents the semantic similarity of words and effectively resolves the synonymy issue. However, LSA ignores the order of words and contextual information. Some researchers have proposed improvements to address these limitations. Peng, X. et al. [41] divided essays into multiple ordered parts, representing each part by the LS-VS model, and combined them into composition vectors in order, thereby adding sequential information into the semantic features. Xu, Y. et al. [38] proposed a Contextualized Latent Semantic Index (CLSI) method that utilizes Weighted Finite-State Transducers (WFST) to extract local contextual information of words in sentences for obtaining feature representation vectors to construct the LS-VS model.

In recent years, deep neural network-based embedding models such as word2vec (word to vector) and BERT, which incorporate sequence features and multidimensional text features, have been widely applied in ACEE feature extraction and scoring tasks. Some studies directly use the one-hot representation of the raw essay texts [50,60]. In contrast, others employ word embedding vectors pre-trained on large-scale corpora to encode essays to obtain more comprehensive semantic information [34,45]. Pretrained language models such as BERT in Chinese [34,45], HBiLSTM [45], and NEZHA [34] have provided convenience for evaluating essays using deep learning methods. Most studies adopted multi-level embedding representations to extract features at multiple levels. Wei, S. and Song, W. et al. [45,60] used a sentence-paragraph two-level neural network model to represent text. At the same time, Li, H. et al. [50] incorporated two levels of Bi-LSTM layers on top of BERT-encoded sentences to obtain a three-level embedding feature representation of sentences, paragraphs, and essays, capturing the text's local and global contextual information.

### 3.2.10. Comparative Analysis of Feature Usage

The utilization of features in ACEE is influenced by factors such as extraction difficulty [8], corpora, and scoring context [61]. Figure 5 illustrates the usage of features reported in the literature, including corpus information for analysis purposes. Embedding-based features include implicit and hybrid features with complex relationships with other specific features. Due to their particularity, we will not conduct a comparative analysis of embedding-based features.

From the overall perspective of feature usage, the frequency of feature utilization can be ranked from high to low: vocabulary, length, topic, grammar, rhetoric, coherence, misspelling, and syntax. This order generally aligns with the difficulty of feature extraction. Length and vocabulary are the most frequently used features for two main reasons. Firstly, these features are often statistical, making them relatively easy to obtain. Secondly, vocabulary and various length-related features are fundamental in distinguishing essay levels in most essay evaluation scenarios in China. Although grammar errors belong to shallow linguistic features, Chinese places more emphasis on semantics than structure, with flexible grammatical structures making extraction challenging. Topic, coherence, and rhetoric features reflect the basic elements of Chinese writing abilities, such as understanding the prompt, organizing the structure, and expressing ideas [62]. Despite the high challenge of extracting these features, they remain crucial for researchers. The complexity of syntax and spelling errors can effectively differentiate the language proficiency of novice learners, playing a vital role in evaluating essays by non-native Chinese language learners or

native speakers in the early grades. However, extracting these features is still challenging, resulting in their relatively limited usage in ACEE research.

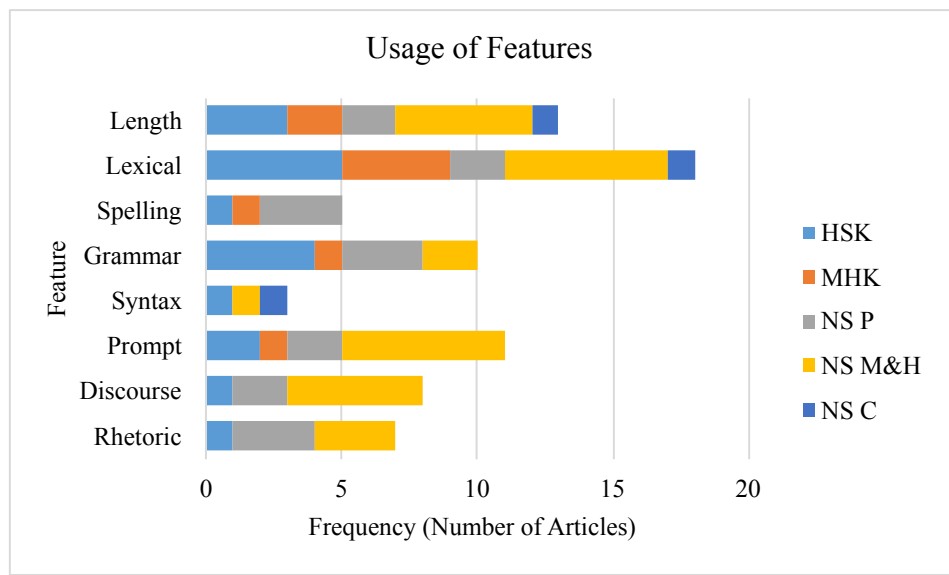

**Figure 5.** Usage of features.

The carefully annotated corpus has significant feature extraction advantages and can serve as training data for feature engineering. Research conducted on the HSK corpus shows a high frequency of grammatical and character feature usage besides vocabulary and length, primarily because the corpus itself has been annotated with this information.

The differences between NS and NNS writing pose unique considerations for automated assessment. Research on HSK and MHK corpora represents the evaluation of Chinese as a second language and foreign language learners' composition, in which the most commonly used features are vocabulary, length, grammar, and topic, emphasizing the use of basic components in the evaluation criteria for second language writing, i.e., learners' general language proficiency [18]. In evaluating compositions written by native Chinese elementary school students, attention is focused on character variation, grammar, and rhetoric, which aligns with the content emphasized in the Chinese language teaching standards [63]. In the evaluation of compositions written by more advanced native language learners, research on discourse, rhetoric, and thematic features predominates, in line with the requirements of the Chinese language curriculum standards [63,64] for the evaluation of intermediate and proficient writing abilities of students in the advanced writing stage.

### 3.3. Scoring Model (RQ3)

The task of scoring has always remained at the forefront of AEE. Researchers have tried to construct scoring models using different machine learning methods, including traditional and deep neural network-based deep learning methods. This article investigates the application of these methods and explores their effectiveness in building scoring models within the context of ACEE.

### 3.3.1. Traditional Machine Learning Approaches

Traditional machine learning methods typically rely on carefully designing, extracting, and selecting features highly relevant to essay scoring. To construct scoring models, these methods utilize conventional regression, classification techniques, or feature similarity.

(1) Traditional classification and regression models

Regression models can predict scores in continuous numerical form. The most commonly used scoring model in AEE is the multiple linear regression model that performs stepwise linear regression analysis on various extracted features to obtain the optimal combination of weighted features (i.e., the most important factors affecting scoring) and establishes a scoring model.

Regression models help predict continuous numerical scores. The most commonly used scoring model is the ACEE's multiple linear regression (MLR) [18,19,33,65]. These models employ stepwise linear regression analysis on extracted features to obtain the optimal combination of weighted features, representing the most influential factors affecting scoring. The scoring model constructed using linear regression demonstrates a high correlation with scores assigned by human raters and possesses strong interpretability. Support vector regression (SVR) [41,57,59] is an extension of SVM, where the hyperplane decision boundaries calculated by SVM are utilized as the regression model to predict essay scores. It is commonly used in AEE for scoring tasks with small corpora. Besides these models, Bayesian regression [36], Bayesian linear ridge regression [62], and gradient boosting decision tree (GBDT) regression models help predict continuous numerical scores. The most commonly used scoring model is the ACEE's multiple linear regression (MLR) [18,19,33,65]. These models employ stepwise linear regression analysis on extracted features to obtain the optimal combination of weighted features, representing the most influential factors affecting scoring. The scoring model constructed using linear regression demonstrates a high correlation with scores assigned by human raters and possesses strong interpretability. Support vector regression (SVR) [41,57,59] is an extension of SVM, where the hyperplane decision boundaries calculated by SVM are utilized as the regression model to predict essay scores. SVR-based methods are commonly used in AEE for scoring tasks with small corpora. Besides these models, Bayesian regression [36], Bayesian linear ridge regression [62], and gradient boosting decision tree (GBDT) [45] have also shown promising performance in ACEE scoring tasks.

For AEE research involving essay scoring as categorical data, classification methods are employed for building scoring models. Ordered logistic regression [35], SVM [38,40,44,66], decision trees [20], and random forests [43] are among the methods used to construct classification models for essay scoring. Due to its nonlinear characters and high-dimensional features, SVM is particularly suitable for constructing scoring models that offer high accuracy without requiring large-scale corpora, so it is a preferred method with which researchers build classification scoring models.

(2) Feature similarity model

Some studies build scoring models by evaluating the similarity of specific features between the target and standard essays of different proficiency levels to assign scores. Latent Semantic Analysis (LSA) is a typical model that scores essays based on semantic similarity by computing the similarity in the latent semantic space between the target essays and standard essays [21]. Other features beyond latent semantics have also been utilized in constructing similarity-based scoring models. Peng, X. et al. [41] computed the probability distribution of vocabulary in each rating category and the weights of vocabulary distribution probabilities in the scoring model, thereby establishing a vocabulary-based scoring model that essentially relies on the similarity of vocabulary distribution probabilities for essay scoring. Moreover, Chang, T. et al. [52,67] measured the similarity in the number of literary semantic primitives between the essays in the standard essays set and the target essays, as well as the similarity in conceptual structure between paragraphs, to assign scores to the target essays.

3.3.2. Deep Learning-Based Scoring Model

The deep learning methods based on DNN can bypass the feature engineering stage by automatically extracting features for essay scoring through hidden layers. These scoring

models employ text embedding as typical input and continuously update feature values through training on a large corpus to achieve optimal predictive results.

Deep learning models generally require a large-scale training corpus and lack interpretability in feature extraction, making it challenging to support feedback and recommendation generation. To address these issues, researchers have employed deep learning methods such as transfer learning, multi-task learning, and fusion of feature engineering to construct scoring models.

(1) Transfer learning

The performance of the scoring model can be affected by variations in writing style, topic, and the author's grade level. Therefore, ACEE research primarily adopts the following two methods to enhance the model's adaptability.

The first method, known as prompt-free or prompt-independent, constructs evaluation models by selecting feature sets that are not affected by thematic factors from multi-topic corpora. However, due to the absence of topic-related features, this method may result in high biases in some scores. Wang, Y. et al. [35] found that the scoring system would overestimate the scores for essays demonstrating outstanding language proficiency while lacking relevance to the given topic.

The second method enables the scoring model to adapt to grade levels and topic ratings by employing transfer learning. Song, W. et al. [60] improved the performance of the DNN model in cross-topic scoring through three stages: pre-training using weakly supervised data and cross-topic data, fine-tuning using target-topic data which needed less than 1000 training essays for one prompt. Wei, S. et al. [45] designed a general scoring model calibration scheme that can adapt to grades and themes. Firstly, a general scoring model was trained using a universal scoring dataset disregarding grade levels and topics. Then, the independent fully connected regression score layers were used to train grade-specific general scoring models for grade-level scoring tasks using essays from different grade levels. Finally, a Bayesian ridge regression model was trained using a small target-topic essay dataset to predict scores for specific-topic essays.

(2) Multi-task Learning

The pre-trained embedded model needs sufficient data for fine-tuning in ACEE scoring tasks. Still, the fine-tuning effect cannot be guaranteed if the target topic corpus is insufficient. To address this issue, Sun, J. et al. [34] employed a multi-task learning approach by incorporating two auxiliary tasks, namely topic prediction and topic matching, into the scoring task. The model learns a shared representation or feature extractor across these tasks by sharing parameters. This method facilitates the model's ability to capture task-specific patterns and shared underlying structures, improving generalization and efficiency, so it effectively resolves multi-topic corpora-trained models' performance limitations when scoring essays on specific topics.

(3) Hybrid Features

Traditional scoring methods based on feature engineering and feature-free deep learning methods have advantages and limitations. However, a hybrid scoring model that combines both methods can achieve both accuracy and interpretability in scoring. In their work, Yuan, S. et al. [47] extracted 35 commonly used statistical features and embedded semantic features represented by LSI using Coh–Metrix. A CNN model was then built to predict scores based on these features to predictive scores. Wei, S. et al. [45] employed various NLP methods to extract shallow statistical features and deep features related to language usage, language expression, discourse anomalies, and text quality. These features were combined with text embedding features obtained through pre-trained BERT. The combined features were fed into a DNN to train a hybrid features scoring model.

### 3.3.3. Evaluation Matric

The performance of the ACEE scoring model is typically evaluated by measuring the extent to which the model replicates human scores. The evaluation methods commonly used in research include Accuracy Rate (AR), Exact Accuracy Rate (EAR), Root Mean Square Error (RMSE), Average Error (AE), Quadratic Weighted Kappa (QWK), and Pearson Correlation Coefficient (PCC). AR and EAR are used to assess the accuracy of the scores. AR represents the rate of score agreement and is sometimes referred to as adjacent agreement, which measures the machine's accuracy within a certain range of score differences (e.g., 10% of the total score range for high school essay grading, which is 6 points). EAR, also called exact agreement, measures the rate at which the model's scores match exactly with the human scores. RMSE and AE quantify the deviation between the model's and human scores. RMSE calculates the root mean square error between predicted and actual scores, while AE measures the average difference between predicted and human scores in the validation set. QWK and PCC are the most commonly used evaluation metrics in the ACEE field, aiming to measure the correlation between human and machine scores. The results of QWK and PCC are usually consistent.

Overall, the usage of evaluation methods from the literature is illustrated in Figure 6, where we can observe that AR and PCC are the most frequently used evaluation methods, while RMSE and AE are less commonly employed in ACEE research. Many studies utilize multiple evaluation methods to provide a more objective and comprehensive evaluation of the models. Among the literature, 17 papers (58.6%) employed two or more evaluation methods. However, only a small portion (14.8%, three articles) evaluate the models from accuracy, score deviation, and correlation perspectives. Among these, most researchers (approximately 52.9% of the articles using multiple evaluation methods) prefer to evaluate the scoring performance of the models through accuracy and the correlation between human and machine scores.

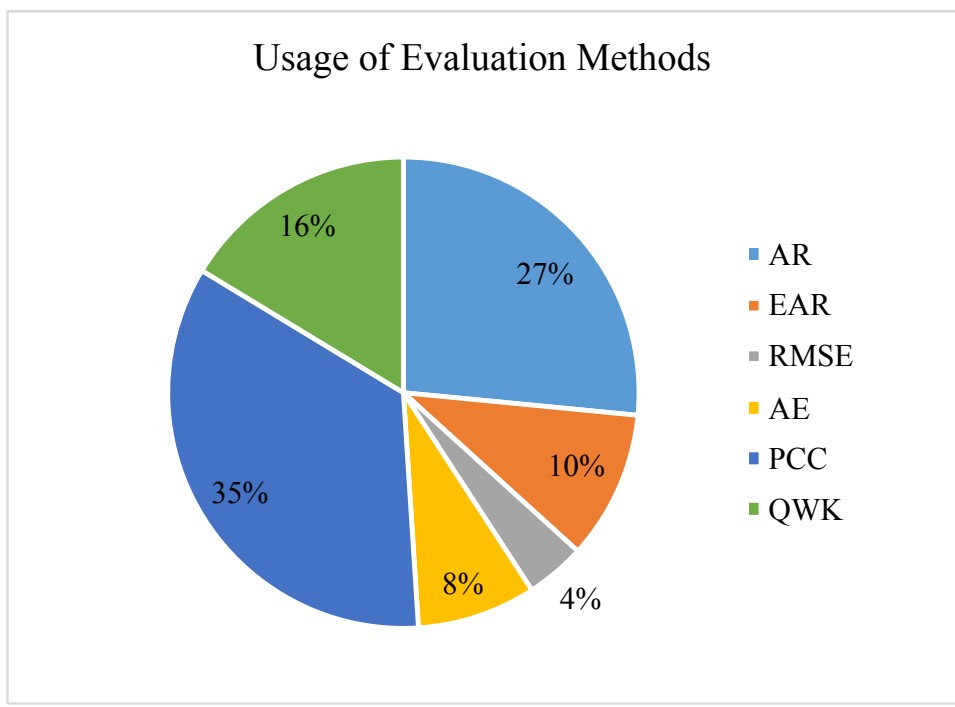

**Figure 6.** Usage of evaluation methods.

### 3.3.4. Comparative Analysis of Scoring Models

The performance of scoring models is closely related to the scoring criteria (relevant to the scoring context), the size of the corpora, the selected features, and the methods used to construct scoring models. Table 3 compares the construction details and performance

of scoring models selected in the literature from the above mentioned perspectives. We classify the scoring models in terms of the corpus sources and scoring models used in the literature, as using the same corpus source implies the same scoring criteria, which enhances the comparability of studies when identical corpora are employed. Moreover, considering that the linguistic level of feature representation has a more significant impact on the choice of scoring methods than specific features, the table categorizes the features as shallow and deep semantic attributes and embedding features. To better compare the performance of scoring models and draw relatively accurate conclusions, we selected the best-performing performance of scoring models from the literature as our evaluation results. Furthermore, due to the substantial differences in the scoring ranges across studies, it is not meaningful to compare the values of RMSE and AE. Thus, these evaluation metrics are not presented in the table. It should be noted that there are some limitations in the comparative analysis presented in Table 3. Firstly, the differences in evaluation results can only serve as reference data due to the differences in the employed corpora and evaluation methods. Secondly, some of the studies selected in this article did not adopt the evaluation methods mentioned earlier (e.g., some studies [42,56] verified the effectiveness of features in scoring by establishing the correspondence between features and essay grades) or did not provide the evaluation results of their models. Therefore, such research findings were not included in the comparative analysis.

Overall, the performance of scoring models is influenced by factors such as the corpus, features, and scoring methods. The relationship between the scoring model's performance and the corpus's size is complex. For example, studies on HSK corpora with around 100 samples [18] and over 10,000 samples [35] show similar PCC values. However, it should be noted that some studies with low performance [21,47] suffer from inadequate samples. DNN models generally need large-scale training corpora. We can see from Table 3 that the relationship between feature selection and the scoring model is not significant except for the embedding features unsuitable for MLR models. Scoring models that utilize multiple hierarchical features tend to achieve a better scoring performance.

As corpora often indicate the scoring context of essays, this study analyzed the performance of scoring models under various corpus backgrounds. Since publicly available corpora are used, studies conducted on HSK corpora have relatively high comparability due to the consistency of the content. We can see from Table 3 that both traditional regression/classification models and DNN models can achieve a human-machine scoring correlation of approximately 0.73 in terms of the PCC values of the models. MHK corpora usually come from different exam times or regions, thus having different content but the same scoring criteria. Studies based on MHK corpora have predominantly used traditional scoring models, focusing more on identifying which features and indicators significantly influence essay scoring in MHK exams. MLR methods have demonstrated a higher human-machine scoring correlation in this corpus. In the context of native speakers' scoring, various scoring models mostly achieve accuracies ranging from 84% to 95%, while there are significant variations in human-machine correlation. Notably, scoring models based on deep learning generally yield PCC and QWK values ranging from 0.60 to 0.66 (except for the model in the study of Yuan, S. et al. [47], which shows a lower PCC value, possibly due to insufficient data and limitations in the selected features), reflecting the current state-of-the-art performance benchmark of deep learning models in ACEE scoring.

**Table 3.** The comparison of ACEE scoring models.

| Reference | Corpus | | Features | | | Scoring Models | Evaluate Results | | | |
|---|---|---|---|---|---|---|---|---|---|---|
| | Category | Size | S | D | E | | AR | EAR | PCC | QWK |
| [18] (2004) | HSK | 100/140 | ✓ | | | MLR | | | TP: 0.767 CP: 0.714 | |
| [33] (2014) | HSK | 1523/9073 | ✓ | | | MLR | TP: 0.84 CP: 0.77 | | | |
| [35] (2021) | HSK | 10,227 | ✓ | ✓ | ✓ | Ordered Logistic Regression | | | 0.734 | 0.714 |
| [34] (2022) | HSK | 8878 | | | ✓ | DNN | | | TP: 0.728 CP: 0.732 | TP:0.678 CP:0.704 |
| [19] (2004) | MHK | 700 | ✓ | | | MLR | | | 0.822 | |
| [37] (2011) | MHK | 994 | | ✓ | | MLR | 0.97 | | 0.92 | |
| [41] (2010) | MHK | 970 | | | ✓ | SVR | | | 0.611 | |
| [44] (2016) | MHK | 16,776 | | | ✓ | SVM | 0.895 | | 0.61 | |
| [38] (2017) | MHK | 12,600 | | | ✓ | SVM | 0.897 | | 0.55 | |
| [40] (2014) | MHK | 16,776 | | ✓ | | SVM | 0.89 | | | |
| [39] (2012) | MHK | 8000 | | ✓ | | Feature Similarity | | | 0.704 | |
| [49] (2020) | NS(M) | 15,000 | | ✓ | | MLR | | 0.78 | | 0.88 |
| [51] (2022) | NS(M) | 10,500 | ✓ | ✓ | | MLR | 0.942 | | | 0.867 |
| [58] (2020) | NS(H) | — | ✓ | ✓ | | MLR | | 0.83 | 0.535 | |
| [62] (2017) | NS(M) | 10,300 | ✓ | ✓ | | BLRR | | | | 0.696 |
| [59] (2018) | NS(M) | 1000/500 | ✓ | ✓ | | SVR | 0.931 | | 0.83 | |
| [57] (2016) | NS(P) | 220 | ✓ | ✓ | | SVR | 0.927 | | 0.82 | |
| [20] (2006) | NS(M) | 693 | ✓ | ✓ | | Decision Tree | 0.91 | 0.48 | | |
| [43] (2019) | NS(P) | 1000 | ✓ | ✓ | | Random Forest | | | | 0.759 |
| [21] (2007) | NS(H) | 202 | | | ✓ | LSA | | | 0.55 | |
| [67] (2008) | NS(M) | 689 | | ✓ | | Feature Similarity | 0.92 | | | |
| [52] (2009) | NS(M) | 689 | | ✓ | | Feature Similarity | 0.84 | 0.37 | | |
| [47] (2020) | NS(M) | 100/107 | ✓ | | ✓ | CNN | 0.759 | | 0.45 | |
| [60] (2020) | NS(M & H) | 85,535 + 3885 | | | ✓ | ARCNN | | | 0.662 | 0.628 |
| [50] (2020) | NS(H) | 300 | | | ✓ | RNN | | | 0.642 | 0.604 |
| [45] (2022) | NS(P) | 8500 + 300~500 | ✓ | ✓ | ✓ | DNN | 0.882 | | 0.636 | |

Note: (1) The meaning of abbreviations: In the Corpus column, NS represents local student compositions, while P, M, H, and C in parentheses represent primary school, middle school, high school, and college, respectively, and the numbers following represent grades. S, D, and E in the Feature columns represent shallow semantic attributes, deep semantic attributes, and embedded features, respectively. In Evaluation Results, TP represents the same topic, while CP represents a cross-topic. (2) In the Corpus Size column, if there is a '/' between two numbers, it indicates that two different corpora are used to train models. The numbers on both sides of '+' indicate pre-training and fine-tuning data sizes.

## 4. Discussion

This section will address the research questions by discussing the gap between current status and desired outcomes.

RQ1: What is the status of construction and usage of corpora in ACEE research?

As the foundation of ACEE research, publicly accessible Chinese essay corpora are currently insufficient. Notably, there are challenges in obtaining local student essay corpora across different educational stages, which restricts the flourishing development of ACEE research. The quantity and quality of research corpora significantly impact research outcomes. In annotating high-quality essay corpora, at least two experienced raters must undergo training or calibration based on the scoring criteria. They then invest considerable time and effort in evaluating the essays. In some studies, issues such as less rigorous manual rating processes and insufficient corpus quantity have affected the accuracy and reliability of the scoring results to varying degrees. Although some research has addressed the scoring issue in small datasets to some extent through pre-trained transfer learning methods, the construction of open essay corpora can provide researchers with high-quality research resources, a unified research platform, and an active research community, according to the experience in English AEE research.

RQ2: What features can ACEE extract for essay evaluation, and what are the key techniques and methods involved in the extraction process?

Feature engineering in ACEE includes feature selection and extraction, essential for constructing scoring models and providing a data basis for specific feedback and suggestions in evaluation results. The development of Chinese NLP technology enables ACEE language features to gradually evolve from statistics and distributions-based shallow quantitative features to deeper language features incorporating more semantic and contextual information. The types of features include length, vocabulary classification, writing style, grammatical errors, topic relevance, discourse structure, rhetoric, etc. Feature extraction methods include statistical-based methods, rule-based methods, data-driven methods, etc. Some text analysis tools can assist in extracting some shallow features, such as the Chinese version Coh-Metrix [47,49,51] and the Language Technology Platform (LTP) [49,51] developed by the Harbin Institute of Technology. However, there are still limitations in the extracted features for essay evaluation tasks. Firstly, the Chinese language has a flexible grammatical structure, which makes it challenging to use syntactic parsers to check for grammar errors. Although deep learning-based sequence labeling techniques become the mainstream Chinese grammar error detection method in recent years, the performance of the leading techniques in this task remains relatively low based on the latest evaluation results from CGED. These techniques are currently only of reference value in practical applications. Secondly, current techniques mostly rely on indirect and macroscopic features to evaluate content quality related to semantics, topics, and discourse. These features perform well in scoring tasks but only support overall comments, which are not conducive to generating specific diagnostic feedback. Thirdly, existing research is lacking in identifying and extracting certain implicit features, such as the quality of rhetoric, the depth of expressing thoughts, and high-level thinking in automated evaluation methods. Lastly, there is a lack of feature extraction for special linguistic styles in Chinese writing, such as classical Chinese, poetry, and other rhythmic genres.

RQ3: What are the key technologies and methods for constructing the ACEE scoring model?

The scoring task is one of the core issues in ACEE research. The methods for constructing scoring models can be broadly classified into traditional machine learning methods and deep learning methods. Traditional machine learning methods require manually extracted features to build models, with commonly employed predictive models including MLP, BLRR, SVR for regression, and SVM and decision trees for classification. MLP models have shown good accuracy in scoring and correlation between human-machine scores across different types of corpora. However, if a high correlation exists among features, using MLP models may not be appropriate. SVR and SVM models have fewer limitations in

feature selection, as they can combine carefully extracted evaluation dimension features with text vector features. Deep neural network models can bypass the complex feature extraction and selection stages but require substantial training data. Deep learning models have implicit feature extraction processes compared to traditional scoring models, making it difficult to explain which factors affect the quality of compositions, leading to doubts and the inability to provide specific feedback information. It is worth noting that, in recent years, the development of deep learning methods in areas such as transfer learning [45,60] and multi-task models [34] has greatly enhanced the scalability of deep neural network models, creating broader opportunities for research in scoring models. Moreover, the integration of feature engineering and deep learning models has been bringing about a brighter prospect for applications.

## 5. Conclusions

This article conducted a systematic literature review of 29 high-quality research works related to ACEE technology published between 2004 (the widely acknowledged year of publication for the first ACEE literature) and 2022. As critical technological issues, the development of corpus construction, feature engineering, and evaluation models collectively contribute to the advancement of ACEE.

In summary, over the past decade, ACEE has made substantial progress in research on text preprocessing, feature extraction, and evaluation models. In recent years, multiple ACEE products have been applied in practice. However, research in this field is still in its early stages, with fragmented studies and fewer systematic or inherited research efforts. Chinese writing exhibits diverse expression patterns and a complex structure, which differ significantly from English in terms of vocabulary construction, sentence organization, and discourse structure. Hence, selecting and extracting features that determine the quality of Chinese essays is inherently more complex. Its application in the field of education is highly promising, and its development requires the integration and advancement of multiple related fields such as Chinese linguistics, pedagogy, and computer science. The ACEE research community and platforms are emerging, calling for broader participation from researchers.

Looking ahead, there are several avenues for future research and development in ACEE:

- Corpus Expansion: To enhance the effectiveness of ACEE models, creating more extensive and diverse Chinese essay corpora is imperative. Researchers should collaborate to build and share such corpora, facilitating the development of more robust ACEE systems.
- Language-specific Challenges: Researchers should address language-specific challenges related to Chinese writing, such as handwriting quality, idiomatic expressions, prosody structure, etc. These aspects require dedicated attention to ensure accurate evaluation.
- Deep Learning and Neural Networks: Given the success of deep learning models in English AEE, further exploration and adaptation of these techniques to Chinese AEE may yield significant improvements.
- Data efficiency: The effectiveness of machine learning technologies in ACEE relies heavily on training data volume and diversity. Building comprehensive corpora that represent diverse writing scenarios is expensive and labor-intensive. Data efficiency enhancement is vital.
- Multidimensional scoring: Multidimensional scoring in ACEE involves assessing various aspects of essays, providing more detailed feedback to students, and aiding educators in targeted teaching. This approach comes with challenges like determining the correlation between features and various scoring dimensions and the interaction between features, as well as model development.
- Stability and generalization: Ensuring the stability and generalization of ACEE scoring models is vital for reliable, unbiased essay evaluation. It must consistently assess

various tasks and student profiles, adapting to evolving educational needs while addressing potential biases. This ensures ACEE's versatility and trustworthiness as an equitable essay assessment tool.

- Collaborative Platforms: The emerging ACEE research community and platforms should actively foster collaboration among researchers. Encouraging knowledge-sharing, joint projects, and creating open-source tools will accelerate progress.

In conclusion, while ACEE has made significant strides, it is clear that there is ample room for growth and collaboration. The challenges that are unique to evaluating Chinese essays demand creative solutions and collective efforts from the research community. By addressing these challenges and embracing future opportunities, ACEE can further evolve to meet the needs of educators, learners, and the broader Chinese language community.

**Author Contributions:** Conceptualization, H.Y. and Y.H.; formal analysis, H.Y. and Y.H.; data curation, Y.H., X.B., H.X. and W.G.; writing—original draft preparation, Y.H.; writing—review and editing, H.Y. and Y.H.; supervision, H.Y.; funding acquisition, H.Y. and W.G. All authors have read and agreed to the published version of the manuscript.

**Funding:** The research is supported by the research fund from the National Natural Science Foundation of China (Grant No. 62067008, No. 11664036, No. 62267008) and the Education Technology Innovation Project in Gansu Province (Grant No. 2021CXZX-218). Additionally, part of this work is also supported by the Science and Technology Program of Gansu Province (Grant No. 21JR7RA117).

**Institutional Review Board Statement:** Not applicable.

**Informed Consent Statement:** Not applicable.

**Data Availability Statement:** Not applicable.

**Conflicts of Interest:** The funders had no role in the design of the study; in the collection, analyses, or interpretation of data; in the writing of the manuscript; or in the decision to publish the results.

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
