# Peer review of "Automatic Essay Evaluation Technologies in Chinese Writing—A Systematic Literature Review"

_applsci, doi:10.3390/app131910737_

Round 1

Reviewer 1 Report

In the introduction, the objective of the work is not clear; the contribution and justification of the theme need to be emphasized.

In the Materials and Methods section, it is recommended to unify the research question, revolving around the work's objective. Highlight the phases of the systematic literature review and clarify which authors were considered for systematization.

In Figure 2, the last stage shows 29 articles, while the text states that there were 30 articles. Which number is correct?

What is a Corpus? I can't identify what this means.

I missed a compendium (output) that shows the answers to the research questions. This output is useful for practical applications of the work.

In the conclusion, highlight the practical and managerial implications for academia and society. Emphasize the study's contribution, limitations, and suggestions for future work.

The English requires revision.

In the introduction, the objective of the work is not clear; the contribution and justification of the theme need to be emphasized.

In the Materials and Methods section, it is recommended to unify the research question, revolving around the work's objective. Highlight the phases of the systematic literature review and clarify which authors were considered for systematization.

In Figure 2, the last stage shows 29 articles, while the text states that there were 30 articles. Which number is correct?

What is a Corpus? I can't identify what this means.

I missed a compendium (output) that shows the answers to the research questions. This output is useful for practical applications of the work.

In the conclusion, highlight the practical and managerial implications for academia and society. Emphasize the study's contribution, limitations, and suggestions for future work.

The English requires revision.

Author Response

Thank you very much for taking the time to review this manuscript. Your insightful feedback has played a pivotal role in enhancing the quality and clarity of our work. We have made revisions to the relevant content based on your comments. 

Please see the attachment for more details.

Reviewer 2 Report

This is a well written and important manuscript. I intend to cite this in my upcoming work. I have only minor suggestions for improvement. First, I think it is important to discuss earlier on the difference between automatically assessing L1 and L2 writers. The authors do note this periodically throughout their paper, but at times it is unclear if the statemtents are referring to L1, L2 or both types of writers. Along these lines it is important to note whether the writings being assessed have been typed or written. Much research (specifically in the L2 field) have illustrated how writing modality can impact what is assessed and what can be interpreted from those assessments. Secondly, as a reader I'd like to hear more of the authors commentary on the approaches in the results and discusions section. The authors save much of this for the conclusion, which is fine, but it feels a bit rushed. These are the only suggestions I have. I look forward to seeing this published. 

Author Response

(The authors gave the same response as above.)

Reviewer 3 Report

The paper discusses the state of Chinese Essay Evaluation (ACEE) technology in education, emphasizing its growing importance and current challenges. It identifies research gaps in corpus construction, feature engineering, and scoring models in ACEE. The passage is well-structured, uses technical language appropriately, and provides insights into the field's future potential. Overall, it offers a concise overview of ACEE research.Below are some my concern:

·         The language used throughout the manuscript needs to be improved

·         Improve the introduction part by making it more concise.

·         More discussion on the current limitations Artificial Intelligence (AI) can be helpful.

·         Add  Results and discussuion session and detailed analysis should be added.

·         Include in conclusion session more furutre suggestions.

Minor editing of English language required

Author Response

(The authors gave the same response as above.)

Round 2

Reviewer 1 Report

article accepted.